# Self-Reported Diet and Health Outcomes of Participants of the CCSVI-Tracking Survey Study

**DOI:** 10.3390/nu13061891

**Published:** 2021-05-31

**Authors:** Patricia Grace-Farfaglia

**Affiliations:** 1Department of Health Science, Rocky Mountain University of Health Professions, Provo, UT 84606, USA; grace-farfagliap@sacredheart.edu; 2Department of Health Science, College of Health Professions, Sacred Heart University, Fairfield, CT 06825, USA

**Keywords:** multiple sclerosis, diet, symptoms, quality of life, EDSS

## Abstract

Of the 1575 participants of the CCSVI-Tracking Survey, 475 patients recorded their quality of life and EDSS outcomes for at least 2 months. Self-reported use of complementary and conventional therapies included diet, use of drug therapy, symptoms, quality of life, and mobility. Analysis included comparing outcomes related to different diets within and between groups. Adherence to the MS diet was not associated with a greater quality of life, less disability, a lower Symptom Score, or faster walking speed compared to other diets. Alternately, the participants from the Mediterranean diet region as a whole (µ = 32.65 (SD = 11.37, SE_M_ = 2.37, *p* = 0.05) had a significantly greater QoL (µ = 60, *p* = 0.05) and a lower MS symptom score, µ = 32.65 (11.37), *p* = 0.0029. A decline of symptoms was observed in all diet groups over 3 months with the most dramatic decline observed in participants from the Eastern Mediterranean diet region. The main effect for the within-subjects factor was significant, *F*(3, 1056) = 55.95, *p* < 0.001, indicating that there were significant differences between the groups.

## 1. Introduction

Multiple sclerosis is a disabling condition that is a demyelinating disease involving immune system attacks on the brain and spinal cord. In 2010, the vascular disease specialist Paulo Zamboni theorized that “chronic cerebrospinal venous insufficiency” (CCSVI) played a role in the pathology of multiple sclerosis (MS) [1]. A patient-reported (PRO) data tracking survey recruited by advocates within their social health networks was developed to improve our knowledge of multiple sclerosis. This hypothesis suggested that intraluminal defects, compression, or hypoplasia in the internal jugular or azygos veins results in venous reflux into the brain of patients with MS, associated with MS brain lesions. His research attracted the attention of patients seeking innovations in treatment, many of whom were willing to obtain this through medical tourism [2].

The CCSVI-Tracking Survey was an initiative of a group of volunteers who met on an early Internet forum on CSSVI. Most of the volunteers suffered from MS and had CCSVI treatment with results that varied from zero to substantial positive effects on MS-related symptoms. The group was fully independent and consulted with medical specialists and social scientists in the development of the survey. There were two types of participants, those who wanted to create a registry of MS patients treated with percutaneous angioplasty of venous percutaneous transluminal angioplasty (PTA), and others who saw research value in tracking health outcomes over time, some for as long as 5 years (2010–2015). In total there were 1421 participants and 917 treatments tracked for a total of 751 months. The data presented in this paper is primarily data from up to 12 months after the procedure. Although this was primarily a study of patients from North America and Europe, participation included pwMS from 12 other countries. 

The current state of MS treatment is somewhat less contentious today than it was in 2010 when the tracking survey began, and researchers acknowledge that there were many lessons learned [3,4,5]. Today the pathogenesis of multiple sclerosis is best viewed through a lens that considers immune, vascular, microbiome, and environmental factors [6,7,8,9,10]. Among neuroscientists, there is general consensus that multiple sclerosis and other neurodegenerative disease can best be understood as a nexus of all of systems, including the microbiome, and their interrelations. The role of diet in the prevention and treatment of MS has gained renewed attention in immunology and neurology. The reader is directed to recent reviews on dietary restriction, fat restriction, paleolithic ketogenic diets, and plant-based dietary patterns and their anti-inflammatory and neuroprotective effects [11,12]. While the use of microbiota and their metabolites as disease markers has been reported, these interactions do not reliably predict the response to dietary intervention [13]. From the literature, it is clear that dietary adequacy for micronutrients, moderation of energy intake, and reductions in animal protein and LCFA consumption alongside an Page: 2 increase in n-3 fatty acids, a plant-based diet rich in polyphenols, and fiber with moderate amounts of seafood are all beneficial elements [14,15,16]. There have been no studies on the effect of diet in the early stages of demyelination. 

The main aim of this paper is to report on the dietary choices and health outcomes of pwMS who volunteered because they received venous percutaneous transluminal angioplasty (PTA). Many diets have been proposed for the management of multiple sclerosis and other neurodegenerative diseases [17,18,19,20]. Australians with MS frequently adopt diets low in fat, sugar, and gluten [21]. In the US, the Swank diet was followed by patients to relieve MS symptoms [22]. When naturopaths were surveyed, 52.4% reported that they recommended that MS patients follow a therapeutic diet [23]. A survey of five Nordic countries reported that between 12.2% and 18.7% of respondents followed a special diet [24]. In the National Health Interview Survey of 2012, only 3% of American adults reported being on any type of special diet (95% CI: 2.8–3.3). Popular MS dietary patterns at the time of this survey and their avoidance similarities are displayed in Table 1. Two longitudinal studies by the Chicago Health and Aging Project (CHAP) have demonstrated that the consumption of unhealthy foods along with a Mediterranean diet pattern may attenuate the diet’s benefits on cognition, and likewise an increase in adherence to a Mediterranean diet reduces the rate of cognitive decline (β = +0.0014 per 1-point increase, SEE = 0.0004, *p* = 0.0004) [25]. PwMS in the Netherlands who adopt the Dutch Healthy Diet-Index have higher QoL, physical, and mental health scores [26]. It is evident that no one diet was practiced internationally by pwMS at the time data collection, but dairy, gluten, grains, and ultra-processed foods were commonly avoided across all published diets with the exception of the Mediterranean and vegan diets.

The research questions that informed the analytic approach taken were as follows:Is there a statistically significant difference in quality of life by diet?Is there a statistically significant difference in MS symptoms by diet?Do EDSS scores differ by diet and/or disease-modifying drug therapy?

## 2. Materials and Methods 

At the conclusion of the longitudinal CCSVI-Tracking Survey in 2015, the database administrator offered the data file to any researcher interested in analyzing and reporting the results. Respondent self-selection for this type of population was necessary due to the rarity of this procedure. The use of previously collected data that was deidentified and presented no risk to the participants is secondary research for which consent is not required [32]. There were three separate files: person, treatment, and monthly self-reported health measures. Of the 1421 participants in the person file, there were 553 that matched to the treatment file, and after excluding those contributing less than 2 months of data, there were 476 cases. Time-point data included baseline (one month prior to the PTA procedure) and months 1 through 4. Although there were several participants who contributed for years, the loss of participants in key groups after the first quarter limited any further analysis. 

Aside from personal characteristics, the variables chosen for analysis were selected for their association between dietary pattern, QoL, and symptom severity. Participants indicated in the past month which category best described their condition during the past month on the following 14 functional areas: Mobility (5 items), Balance (8), Coordination (4), Muscle (4), Bladder/Bowel (5), Sensory (6), Speech (5), Cognitive (4), Brain Fog (3), Dizziness (2), Visual (6), Fatigue (4), Temperature (2), MS Attacks, and quality of life (QoL) (1). Scores ranged from 0 (Never) to 4 (Always) with the exception of QoL which was measured on a scale of 0 to 100. The subscale scores were summed into a composite symptom score average. Respondents indicated if the response was based on a professional evaluation for Mobility, Balance, and Vision; for the most part, they were self-reported.

Data reported included sex, year of birth, date of procedure, country, years since diagnosis, MS diet (Never, Only before treatment, Before and after treatment, Only after treatment), Quality of Life (100 being perfect, 0 absence of MS symptoms), Expanded Disability Status Scale (EDSS), MS type, number of PTA treatments, and 6-Meter Timed Walk (6MTWT). The EQ-VAS is a visual analogue scale used to measure quality of life (QoL), anchored by values between 0 (Rock Bottom) and 100 (Perfect, no MS symptoms). The EDSS is an ordinal clinical disability rating scale ranging from 0 (normal neurologic examination) to 10 (death due to MS) in half-point increments. The 6-m walking test was used to assess walking speed in meters per second over a short distance and was part of the overall Symptom Score. 

All statistical analyses were performed using Intellectus Statistics™ (Clearwater, FL, USA) [33]. Descriptive statistics describe the characteristics of the initial cohort at baseline. Data distribution was analyzed for normality, and the transformation of statistics for extreme values was performed. ANOVA with post hoc tests was conducted to determine the paired differences between diet groups.

Before analysis, the normality of the distribution scores for the dependent variables was determined. The Kolmogorov–Smirnov test result was non-significant (*p* > 0.05), but the actual shape of the normal probability plots suggested a reasonably straight line, which indicated a normal distribution of dependent variables. The following sociodemographic characteristics of the respondents were analyzed (Table 2): gender, age group, EDSS, MS diet, and region. Data analysis covers descriptive analysis, the independent-samples *t*-test, the one-way analysis of variance (ANOVA), linear regression, and the multivariate analysis of variance (MANOVA). Statistically significant differences occur if the *p* value is less than 0.05. A categorical independent variable of Diet was recoded to identify MS diet and Western and Mediterranean regional diets.

## 3. Results

Summary statistics were calculated for each interval and ratio variable. Frequencies and percentages were calculated for each nominal and ordinal variable. The most frequently observed sex was female (*n* = 384, 67%). Ages were placed into 3 categories (Under 40, 40–59, and 60 and above) (Table 2). The mean for age was 47.18 (SD = 9.53, *n* = 476). The most frequently identified country of residence was Canada (*n* = 206, 36%), with North America being the largest regional group (57.72%) (Figure 1). Number of PTA treatments had an average of 1.14 (SD = 0.52, SE_M_ = 0.02). Frequencies and percentages are presented in Table 2.

The majority of participants came from North America (*n* = 264, 55.46%) and Europe (*n* = 200, 42.02%) (Figure 1). Personal data did not include education, income information, or comorbidities. Level of disability ranged from moderate to severe disease by the EDSS scale.

### 3.1. Demographic and Clinical Characteristics of Participants

The baseline descriptive data is shown in Table 3. The mean of QoL was 46.59 (SD = 24.63, SE_M_ = 1.13, Mdn = 50.00). The mean baseline mobility measure was the number of seconds taken to walk 6 m (6MTWT) was 5.46 (SD = 3.94, Mdn = 6.50, SE_M_ = 0.18). The mean EDSS of 4.96 (Mdn = 5.50) indicates that the average participant was in the moderate disability category. 

#### 3.1.1. Dietary Patterns and Their Relationship to Health Characteristics and Outcomes 

Participants registered for the survey prior to PTA treatment, and at that time, they were asked if they had followed an MS diet within the past 12 months for more than 50% of the time. Following treatment, participants were asked if they were following the MS diet more than 50% of the time in the past 4 weeks. The responses were recoded into two groups: MSDiet (“MS diet, same as before” and “MS diet since treatment”) and “Western Pattern Diet” groups (Never or Before treatment) for some analyses. To determine whether any one EDSS type was more likely to adopt the MS diet a chi-square test of independence was conducted to examine whether MSDiet (4 levels) and MS Type (5 types) were independent. Results of the chi-square test were significant (α = 0.05, χ^2^(12) = 24.70, *p* = 0.016), suggesting that MSDiet and MSType are related to one another (Table 4). Due to small sample sizes, the Fisher exact test was conducted, and the result was significant (α = 0.05, *p* = 0.011), supporting the chi-square results. The frequencies suggest that participants were more likely to adhere to an MS diet if they had relapsing-remitting disease (RR), and to adopt it after PTA treatment if they had primary progressive disease (PP). MS diet adopters were solely from the US, Canada, and Australia.

Participants in the traditional Mediterranean diet region were not coded as Western Pattern diet, yet food practices have been undergoing changes with a shift to consuming more Westernized or ultra-processed foods [25]. A review of the literature identified regional diet pattern differences in the Mediterranean region [34,35,36,37]. Participants were then coded into one of four groups: MS Diet, Western Pattern Diet, and two Mediterranean diet groups: Eastern (Greece, Turkey) and Western (Spain, Italy, France). The Western pattern diet (WPD) group included participants from Northern Europe, North America, South America, and Australia who responded “never” or in the “past” to the MS diet question. Summary statistics by diet group are found in Table 4. The Eastern Mediterranean diet group had the lowest EDSS and Symptom Scores, and the highest QoL. Surprisingly, the Eastern Mediterranean group (EMed) had the slowest average walking speed for the 6-metre timed walking test speed (6MTWT m/sec). To determine whether walking speed was significantly different, the Kruskal–Wallis Rank sum test results produced significant differences by diet group, χ^2^(3) = 9.21, *p* = 0.027. Pairwise comparisons were examined, and the results of the multiple comparisons indicated significant differences based on an alpha value of 0.05.

There were 36 total members in the combined Mediterranean Diet grouping and 440 in the Other category. A two-tailed independent samples *t*-test was conducted to examine whether the mean of the EDSS baseline was significantly different between the categories of MedDiet and Other. The two-tailed independent samples *t*-test was significant (α = 0.05, t(474) = 2.19, *p* = 0.029) (Table 5). A two-tailed Mann–Whitney two-sample rank-sum test was conducted to examine whether there were significant differences in EDSS baseline between the levels of MedDiet or Other diets. The results of the two-tailed Mann–Whitney U test were significant (α= 0.05, U = 9550, z = −2.07, *p* = 0.039), suggesting that the mean of 4.24 for MedDiet demonstrated a lower level of disability than the mean of 5.0 for the Other group (interquartile range (IQR): 1.5–6.0). 

For EDSS, the difference between a median of 4.00 for MedDiet and 5.50 for Other groups met the MCID criteria of a 1.0. The analysis of the difference for QoL between the combined Mediterranean diet groups used a two-tailed Mann–Whitney U test, and it was not significant (α 0.05, U = 7132, z = −1.00, *p* = 0.316). For the Symptom Score analysis, Welch’s *t*-test was used, as it is more reliable when the two samples have unequal variances and unequal sample sizes [38]. The result of the *t*-test confirmed the initial results (α= 0.05, t(50.79) = 3.04, *p* = 0.004), indicating that the MedDiet group had significantly fewer MS symptoms. The small group (*n* = 7) from the Eastern Mediterranean experienced a significant decline in disability over that 4-month time period, t(249) = 3.78, *p* < 0.001 (Figure 2). The ANOVA for the QoL for diet groups was not significant, F(3, 335) = 0.94, *p* = 0.420, but post hoc analysis showed that overtime the QoL for the MSDiet category was significantly greater than baseline by quarter four, t(335) = −4.29, *p* < 0.001, and QoL was significantly lower for the WPD group, t(335) = −7.24, *p* < 0.001 (Figure 3).

#### 3.1.2. Improvement in EDSS and Symptom Scores in MSDiet Group

The MSDiet responses were recoded into three levels: “Yes, started since treatment”, “Yes, same as before treatment”, and “Never/Before”. A one-way repeated measures ANOVA was conducted to compare the effect of MSDiet levels on QoL over time. For participants who did not adopt an MS diet, QoL significantly declined over time, *F*(3, 729) = 45.41, *p* < 0.001. The QoL for those that abandoned the MS diet before PTA treatment had a significant decline from baseline, *F*(3, 36) = 5.45, *p* = 0.016. Those who adopted the MS Diet after treatment had a significant increase in QoL initially, *F*(3, 72) = 14.55, *p* < 0.001, but group means declined by month 4. The participants that adhered to the MS Diet before and after treatment also reported improvement in QoL and maintained this over the 4-month period, *F*(3, 243) = 13.76, *p* < 0.001. EDSS was significantly greater (*p* = 0.029) in the MSDiet. The mean symptoms were fewer in the MSDiet group (*p* = 0.004). The post-treatment QoL was initially lower but then recovered for early MS Diet adopters that continued dietary adherence (Table 6).

The levels of MS Diet therapy, Before and/or After, were recoded into the MSDiet group and Other. There were 335 observations in the Other diets and 141 observations in the MSDiet group at baseline. For QoL baseline and MSDiet, the two-tailed independent samples *t*-test (0.05, t(474) = 0.00, *p* = 0.999) and the two-tailed Mann–Whitney U test (U = 23,423.5, z = −0.14, *p* = 0.886) were not significant (Table 7). 

#### 3.1.3. No Significant Differences in EDSS between Western Diet Group and Others

There was no significant difference for EDSS between Western Pattern and Other diet groups (Mann–Whitney U test (α = 0.05, U = 23130, z = −1.83, *p* = 0.068.). There was no significant difference in QoL between WPD and other groups (α = 0.05, F(1, 474)t = 0.1, *p* = 0.92) or Symptom Score (α = 0.05, F(1, 474) = 1.18, *p* = 0.277) detected by the independent samples *t*-test (Table 8).

#### 3.1.4. Use of Disease-Modifying Therapies in MedDiet Group 

The chi-square test of independence was conducted to examine whether MS diet adoption and drug therapies were independent. Due to the large number of categories of diet and drug use, Monte Carlo simulations were used to calculate the *p*-value instead of the exact *p*-value. The results of the Fisher exact test were significant based on an alpha value of 0.05, *p* = 0.039, suggesting that adoption of diet and disease-modifying therapy are related (Table 9). MS Diet adopters were more likely to manage their disease with drug therapy at some point in the last 12 months. Participants who adopted the MS diet did not abandon conventional treatment, but instead tried different modalities concurrently. 

#### 3.1.5. MedDiet Associated with Lower Symptom Scores and Higher Baseline QoL 

A hierarchical regression analysis of average Symptoms Score, MedDiet, and age was conducted. Each step in the hierarchical regression was compared to the previous step using F-tests. The Yes category of MedDiet significantly predicted Symptom Score, *B* = −7.96, t(405) = −2.77, *p* = 0.006. Based on this sample, this suggests that moving from the No to Yes category of MedDiet will decrease the mean value of symptom score by 7.96 units on average. Age did not significantly predict symptoms, *B* = 0.15, t(405) = 1.76, *p* = 0.079 (Table 10). The results of a linear regression model for prediction of QoL by MedDiet were significant, *F*(1451) = 9.16, *p* = 0.003, R^2^ = 0.02, indicating that approximately 2% of the variance in QoL is explainable by MedDiet. The MedDiet significantly predicted QoL at baseline, *B* = 12.95, t(451) = 3.03, *p* = 0.003. This suggests that in this sample, the adoption of a MedDiet would increase the mean value of QoL by 12.95 units on average.

Participants in the Mediterranean region on average were younger and had a milder disease than those from other regions (Figure 4). The average age in the MedDiet group was 45.41 (SD = 9.18, Mdn = 46.00) compared to 47.93 (SD = 9.60, Mdn = 48.00) in the Other group.

#### 3.1.6. Symptom Remittance Greatest in EMED Group

Experiencing periods where symptoms are less severe but do not completely cease is a characteristic of multiple sclerosis. Remitting symptoms were observed in all diet groups after the PTA procedure. The most dramatic decline in the Symptom Score was seen in the EMED group, followed by the MS diet, WMED, and finally the WPD. The main effect for each diet group was significant, *F*(3, 352) = 3.72, *p* = 0.012, indicating that there were significant differences in symptom average at baseline, month 1, month 2, and month 3. The main effect for the within-subjects factor was significant, *F*(3, 1056) = 55.95, *p* < 0.001, indicating there were significant differences between the groups. Table 11 and Table 12 present the mixed model ANOVA results and the contrasts for the average symptom score by month with within-factor diet groups. 

## 4. Discussion

The results of this study suggest that the influence of the Mediterranean diet had a significant and positive effect on the quality of life for the respondents in the CCSVI-Tracking Study. This is not surprising, given the HELIAD-linked lifestyle patterns of diet, physical activity, sleep, and social/intellectual pursuits to cognitive health in the general population [39]. However, the effect of the MedDiet alone explained only 2% of the variation in QoL. The degree of disability as measured by the EDSS score at baseline was the lowest in the small Eastern Mediterranean sample and showed further improvement over the four subsequent quarters. The habitual foods in this region include seafood, legumes, Greek-style yogurt, fermented flat bread, homemade cheese, grape leaves and vines, olives and olive oil, and a variety of seasonal fruits and vegetables [34]. As diet habits shift to consuming more saturated fats, sugars, and calories, the gut microbiota composition and host metabolism result in oxidative stress and inflammation [40,41]. 

The MSDiet group, primarily from North America and Europe, was more likely to have advanced disease and to adopt this diet after the venoplasty procedure. Unfortunately, the bar was set low by the survey authors when they defined diet compliance at 50% of the time. The diet recommendations for multiple sclerosis have evolved overtime with the addition of oily fish and other nutrient dense foods, while discouraging all gluten containing grains and dairy [42]. In this cohort, the MSDiet did not reduce symptoms as well as the MedDiet. Animal studies on how methionine intake modulates oxidative stress has yielded a new dietary direction for the treatment of neurodegenerative disease and longevity [43]. Previously, an international study of dietary factors in pwMS found that a high quality diet resulted in lower rates of relapse and better health-related quality of life (HRQoL) [14]. Recently it has been demonstrated in a mouse model that dietary methionine availability is linked to the control of autoimmune T cell activation through a mechanism silencing the MAT2A gene and reducing the total pool of S-adenosyl-methionine (SAM), which is required for the methylation of H3K4 [44,45]. High intakes of methionine-rich animal proteins result in Th17 cell proliferation and cytokine production potentially leading to inflammation and autoimmune disease. Plant-based dietary patterns like Mediterranean, Japanese, and vegan diets tend to be lower in methionine [46]. 

Current guidelines by the European Society of Clinical Nutrition and Metabolism (ESPEN) do not include a recommendation for any diet pattern that extends survival for pwMS but supports dietary restrictions of saturated fats, increasing polyunsaturated fats, and obtaining adequate vitamin D [47]. Whole fruits and vegetables have antioxidant and anti-inflammatory properties and should be part of the core recommendations for pwMS. Researchers have identified disease-specific associations between long-term diet patterns, microbial clusters, and biomarkers [48]. Regular consumption of fish, nuts, vegetables, and cereals improves the generation of short-chain fatty acids (SCFA) known to lower inflammation through the maintenance of gut barrier function. The PREDIMED trial demonstrated that habitual intake of the MedDiet results in higher counts of SCFA-producing bifidobacteria and Bacteroidetes [40].

This survey and its data have some limitations. Firstly, it is a longitudinal study of self-reported data, so causal inferences cannot be drawn; the author has only described associations and changes over a period of 4 to 12 months. Secondly, the sample consisted of a unique group of pwMS who were early adopters of the PTA procedure, as well as other complementary approaches such as low dose naltrexone and inclined bed therapy. This population has a higher risk of cardiovascular disease [49,50] and therefore the results cannot be extrapolated to the general multiple sclerosis population. The reputation as a “liberation therapy” could be a confounding factor as it may be viewed as a placebo effect. For this analysis, it was a factor that was common to all in an international cohort. 

Diet has an important impact on overall health. Increasingly, nutrition researchers find it difficult to identify subjects who have no or limited exposure to ultra-processed foods, and recently some have concluded that eating processed foods along with plant-based foods eliminates their health benefits [25]. Individuals with autoimmune conditions report cognitive improvement with gluten elimination from the diet [51]. This could explain the adherence to the MS Diet in our sample, but when the cognitive items in the Symptom Score were analyzed separately, no significant differences between groups were found. Those living in the Mediterranean region had significant benefits in lower symptoms and positive change in EDSS. It is interesting to note that no participant from the Mediterranean region adopted the MS Diet. The Eastern Mediterranean subgroup, who exhibited greater resiliency post-PTA hopefully retains many of their traditional social activities and dietary patterns to maintain their cognitive health. 

## 5. Conclusions

The Western diet was associated with greater disability and a lower quality of life in an international cohort of individuals with MS and CCSVI. There was no statistical difference between outcomes for the MS and Western Pattern diets. There was a continuum of impact on QoL, disease severity, and symptoms in the Mediterranean group, with those in the Eastern countries having better outcomes. 

## Figures and Tables

**Figure 1 nutrients-13-01891-f001:**
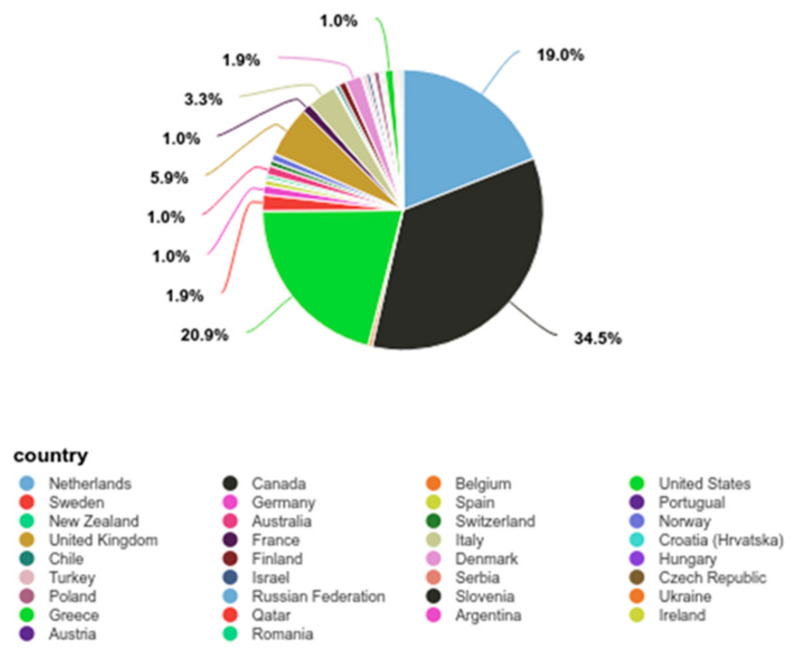
Participation by country.

**Figure 2 nutrients-13-01891-f002:**
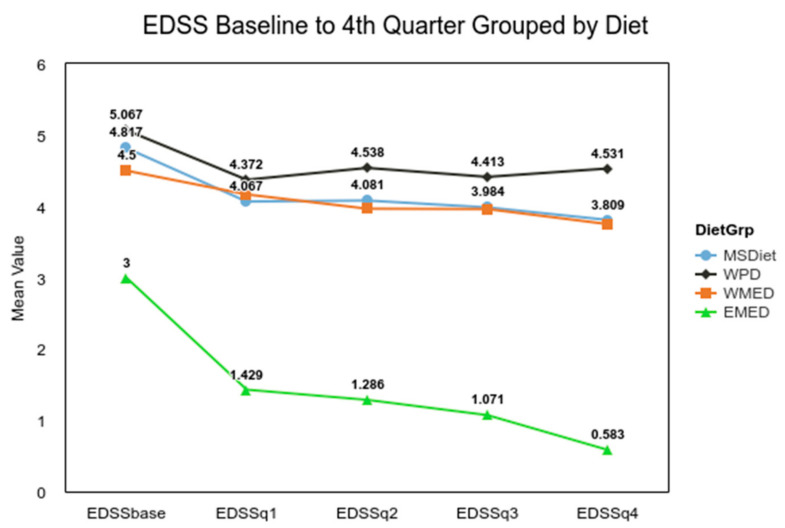
EDSS Baseline to 4th quarter grouped by diet. Note. *n* = 478 Baseline, Quarter 1 *n* = 421, Quarter 2 *n* = 336, Quarter 3 *n* = 284 and Quarter 4 *n* = 253. Q4. MS Diet = MSDiet, Western Pattern Diet = WPD, Western Mediterranean = WPD, and Eastern Mediterranean Diet = EMED. EDSS differences between groups were significant, *F*(3, 249) = 6.44, *p* < 0.001.

**Figure 3 nutrients-13-01891-f003:**
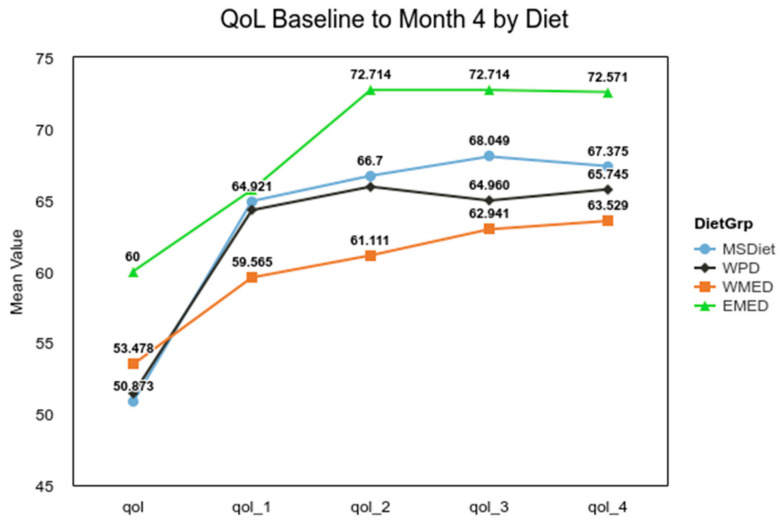
QoL by diet pattern over 4 months. Note. MS Diet = MSDiet, Western Pattern Diet = WPD, Western Mediterranean = WMED, and Eastern Mediterranean Diet = EMED.

**Figure 4 nutrients-13-01891-f004:**
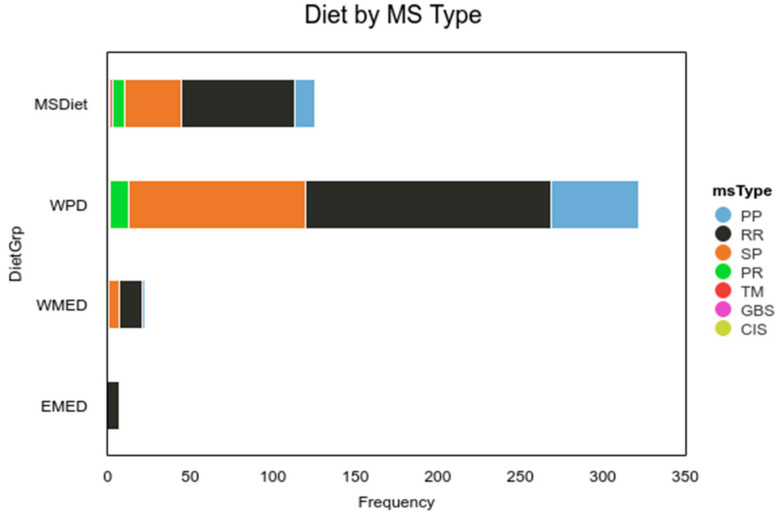
Diet by MS type. Note. MS Diet = MSDiet, Western Pattern Diet = WPD, Western Mediterranean =WPD, and Eastern Mediterranean Diet = EMED. EDSS categories: PPMS = Primary Progressive, SPMS = Secondary Progressive, RRMS= Relapsing-Remitting, PRMS = Progressive Relapsing, TM = Transverse Myelitis, GBS = Guillain-Barre Syndrome, CIS=Clinically Isolated Syndrome.

**Table 1 nutrients-13-01891-t001:** Diet patterns found on MS social media sites and foods that are avoided.

Avoid/Limit:	Gluten-Free	Dairy-Free	Grains Legumes	Fat/Oils	Sugar	Animal-Derived Foods	Processed Foods
Best Bet	X	X	X		X	Red meat	X
Wahls/Paleo	X	X	X		X		X
Swank	X	X	X	X	X	Red meat	X
Paleo	X	X	X		X		X
Swank + Paleo	X	X	X	X	X		X
McDougall/Vegan		X		X	X	X	X
Mediterranean					X		X

Note: Diets of the time period of the survey [18,19,27,28,29,30,31].

**Table 2 nutrients-13-01891-t002:** Demographic and clinical variables of the study participants.

Category	Number	Percent
Sex		
Male	162	34.03
Female	314	65.97
Age at enrollment		
Under 40	113	23.74
40–59	320	67.23
60 and above	43	9.03
MS Types ^1^		
PPMS	76	13.19
SPMS	181	31.42
RRMS	288	50
PRMS	25	4.34
Missing	6	1.04
Region		
North America	264	55.46
Europe	200	42.02
Australia/NZ	8	1.68
South America	3	0.63
Middle East	1	0.21

Note. ^1^ The MS Types categories: PPMS = Primary Progressive, SPMS = Secondary Progressive, RRMS = Relapsing-Remitting, PRMS = Progressive Relapsing.

**Table 3 nutrients-13-01891-t003:** Summary statistics for Age, Age at Diagnosis, 6MTWT, QoL, and Symptom Score.

Variable	*n*	M	SD	Mdn	SE_M_	Min	Max	Skewness	Kurtosis
Age ^1^	476	47.18	9.53	48.00	0.44	22.00	78.00	−0.10	−0.13
Age at Diagnosis	476	36.64	9.70	36.00	0.44	6.00	69.00	0.25	−0.04
EDSS baseline ^2^	476	4.95	2.03	5.50	0.09	0.00	9.00	−0.46	−0.71
6MTWT baseline ^3^	384	6.78	3.29	8.00	0.16	1.00	10.00	−0.68	−0.98
QoL baseline	476	51.49	21.52	50.00	0.99	0.00	100.00	−0.13	−0.55
Symptoms ^4^	476	38.58	16.81	37.00	0.77	0.00	86.00	0.21	−0.15

Note. ^1^ Age was calculated as birth year minus treatment date; ^2^ The Expanded Disability Status Scale (EDSS) scores corresponding to each severity are as follows: Asymptomatic (0); Mild (0 < EDSS ≤ 3.5); Moderate (3.5 < EDSS ≤ 6.5); Severe (6.5 < EDSS ≤ 9.5); ^3^ Outliers removed; ^4^ Symptoms are the average of 16 domain scales. ^3^ Higher 6-Metre Timed Walking Test (6MTWT) scores indicate slower walking speed, EDSS > 6.00 removed from the analysis, as they are restricted to wheelchair or bed.

**Table 4 nutrients-13-01891-t004:** Summary statistics table for EDSS, QoL, Symptom Score, and 6MTWT by diet group.

Variable	*n*	M	SD	SE_M_	Mdn	Min	Max	Skewness	Kurtosis
EDSS baseline									
EMed	7	3.00	1.68	0.64	2.5	1.5	6	0.85	−0.64
WMed	23	4.50	2.11	0.44	4.5	1	8	−0.01	−1.26
MSDiet	126	4.82	2.11	0.19	5.0	0	9	−0.34	−0.88
WPD	322	5.07	1.99	0.11	6.0	0	9	−0.58	−0.49
QoL									
EMed	7	60.00	17.32	6.55	60	40	80	0	−1.57
WMed	23	53.48	17.74	3.70	50	10	90	0.01	0.27
MSDiet	126	50.87	22.06	1.96	50	10	90	−0.24	−0.74
WPD	322	51.40	21.67	1.21	50	0	100	−0.07	−0.53
Symptom Score baseline									
EMed	7	32.14	9.91	3.74	31	19	51	0.79	0.18
WMed	23	32.65	11.37	2.37	34	10	52	−0.05	−1.05
MSDiet	126	39.26	16.65	1.48	38	0	86	0.28	−0.08
WPD	322	38.86	17.31	0.96	38	0	90	0.13	−0.26
6MTWT (m/sec)									
EMed	7	9.00	1.91	0.72	10.0	5.0	10.0	−1.54	0.79
WMed	16	7.69	3.42	0.85	9.5	1.0	10.0	−1.20	−0.23
MSDiet	102	7.09	3.19	0.32	8.5	1.0	10.0	−0.94	−0.75
WPD	103	7.02	3.25	0.32	8.0	1.0	10.0	0.74	−0.95

Note. Categories of diet: WMed = Western Mediterranean, EMed = Eastern Mediterranean, MSDiet, and WPD = Western Pattern Diet. Symptom Score: 14 items; (0) Never to (4) Almost always; QoL (0–100). Expanded Disability Status Scale (EDSS) = Asymptomatic (0); Mild (0 < EDSS ≤ 3.5); Moderate (3.5 < EDSS ≤ 6.5); Severe (6.5 < EDSS ≤ 9.5). Symptom Score lower = less disability. 6-Metre Timed walking Test (6MTWT) (m per sec) higher score = greater speed, outliers and EDSS > 6.0 removed from analysis.

**Table 5 nutrients-13-01891-t005:** Two-tailed independent samples *t*-test for QoL, EDSS, and Symptoms by MedDiet and Other.

	Other		MedDiet				
Variable	M	SD	M	SD	*t*	*p*	*d*
QoL	51.18	21.81	55.28	17.48	−1.1	0.273	0.21
EDSS baseline	5	2.02	4.24	2.08	2.19	0.029 ^†^	0.37
Symptoms	39.03	17.14	33.03	10.78	3.04	0.004 *	0.42

Note. * *p* < 0.01, ^†^
*p* < 0.05. N = 476. Degrees of freedom for the t-statistic = 50.79. *d* = Cohen’s *d.* MedDiet = Mediterranean Diet., EDSS = Expanded Disability Status Scale.

**Table 6 nutrients-13-01891-t006:** Summary statistics of QoL by MS Diet levels.

Variable	M	SD	*n*	SE_M_	Min	Max	Skewness	Kurtosis
QoL baseline								
Yes, started since treatment	53.55	23.74	31	4.26	0.00	90.00	−0.42	−0.37
Yes, same as before treatment	47.12	24.72	125	2.21	0.00	90.00	−0.28	−0.74
Never/Before	43.12	25.68	320	1.44	−1.00	90.00	−0.17	−0.82
QoL month 1								
Yes, started since treatment	68.39	16.14	31	2.90	40.00	90.00	−0.22	−1.05
Yes, same as before treatment	65.60	19.57	125	1.75	10.00	100.00	−0.43	−0.47
Never or Before	64.00	20.94	320	1.17	10.00	100.00	−0.48	−0.41
QoL month 2								
Yes, started since treatment	63.33	23.09	3	13.33	50.00	90.00	0.71	−1.50
Yes, same as before treatment	65.29	22.39	17	5.43	10.00	90.00	−1.00	0.35
Never/Before	65.56	20.62	72	2.43	10.00	100.00	−0.76	0.15
QoL month 3								
Yes, started since treatment	50.00	0.00	2	0.00	50.00	50.00	-	-
Yes, same as before treatment	71.43	14.06	14	3.76	50.00	90.00	−0.09	−1.21
Never/Before	63.87	22.35	62	2.84	10.00	100.00	−0.57	−0.43
QoL month 4								
Yes, started since treatment	50.00	0.00	2	0.00	50.00	50.00	-	-
Yes, same as before treatment	70.00	17.97	14	4.80	30.00	90.00	−1.07	0.10
Never/Before	64.78	20.77	67	2.54	10.00	90.00	−0.72	0.00

Note. ‘-’ indicates the statistic is undefined due to constant data or an insufficient sample size.

**Table 7 nutrients-13-01891-t007:** Two-tailed independent samples *t*-test for QoL, EDSS, and Symptoms at baseline MSDiet/Other.

	Other		MSDiet				
Variable ^1^	M	SD	M	SD	*t*	*p*	*d*
QoL b	51.49	21.32	51.49	22.07	0	0.999	0
EDSS b	5.02	1.98	4.77	2.15	1.19	0.235	0.12
Symptoms b	38.6	17.02	38.52	16.38	0.04	0.966	0

Note. N = 476, ^1^ baseline data. α = 0.05. Degrees of freedom for the t-statistic = 474. *d* = Cohen’s *d*.

**Table 8 nutrients-13-01891-t008:** Two-tailed independent samples *t*-test for outcome variables by Western Pattern Diet (WPD).

	Other		Western				
Variable ^1^	M	SD	M	SD	*t*	*p*	*d*
QoL b	51.63	21.64	51.42	21.5	0.1	0.92	0.01
EDSS b	4.68	2.12	5.08	1.96	−2.08	0.038 ^†^	0.20
Symptoms b	37.55	16.1	39.13	17.18	−0.98	0.33	0.09

Note. N = 476. ^1^ b = baseline. ^†^
*p* < 0.05. Degrees of freedom for the *t*-statistic = 474. *d* = Cohen’s *d.*

**Table 9 nutrients-13-01891-t009:** Observed [Expected]. Frequencies of MS diet and disease-modifying drug therapy.

	Diet				
DMT	Never	Before	Before/After	After	*p*
Never	205[197.50]	15[10.70]	69[78.59]	20[22.22]	0.039
Before	146[148.28]	7[8.03]	58[59.00]	21[16.68]	
Before/After	119[120.16]	4[6.51]	57[47.81]	8[13.52]	
After	10[14.06]	0[0.76]	7[5.60]	5[1.58]	

**Table 10 nutrients-13-01891-t010:** Summary of hierarchical regression analysis for MedDiet predicting Symptom Score.

Variable	*B*	*SE*	95% CI	β	*t*	*p*
Step 1						
(Intercept)	41.39	0.86	[39.71, 43.08]	0.00	48.37	<0.001
MedDiet (Yes)	−9.16	2.80	[−14.67, −3.65]	−0.16	−3.27	0.001
Step 2						
(Intercept)	34.15	4.21	[25.88, 42.42]	0.00	8.12	<0.001
MedDiet (Yes)	−7.96	2.88	[−13.62, −2.30]	−0.14	−2.77	0.006
Age	0.15	0.09	[−0.02, 0.32]	0.09	1.76	<0.079

Note. *F* (1, 406) = 10.67, *p* = 0.001, ΔR2 = 0.03.

**Table 11 nutrients-13-01891-t011:** Mixed model ANOVA average Symptom Score by month with within-factor diet groups.

Source	*df*	*SS*	*MS*	*F*	*p*	η_p_^2^
Between-Subjects						
Diet Group	3	8039.60	2679.87	3.72	0.012	0.03
Residuals	352	253,418.78	719.94			
Within-Subjects						
Within Factor	3	11,382.80	3794.27	55.95	<0.001	0.14
DietGrp:Within.Factor	9	857.79	95.31	1.41	0.231	0.01
Residuals	1056	71,615.34	67.82			

**Table 12 nutrients-13-01891-t012:** The marginal means contrasts for each combination of within-subject variables for the mixed model ANOVA.

Contrast	Difference	*SE*	*df*	*t*	*p*
DietGrp|EMED					
SympAvgb–SympAvgm1	21.86	5.59	352	3.91	<0.001
SympAvgb–SympAvgm2	23.29	5.95	352	3.91	<0.001
SympAvgb–SympAvgm3	24.43	6.03	352	4.05	<0.001
SympAvgm1–SympAvgm2	1.43	2.01	352	0.71	0.893
SympAvgm1–SympAvgm3	2.57	2.57	352	1.00	0.749
SympAvgm2–SympAvgm3	1.14	1.63	352	0.70	0.897
DietGrp|MSDiet					
SympAvgb–SympAvgm1	11.77	1.67	352	7.03	<0.001
SympAvgb–SympAvgm2	13.90	1.78	352	7.80	<0.001
SympAvgb–SympAvgm3	13.68	1.81	352	7.57	<0.001
SympAvgm1–SympAvgm2	2.13	0.60	352	3.54	0.003
SympAvgm1–SympAvgm3	1.91	0.77	352	2.48	0.064
SympAvgm2–SympAvgm3	−0.22	0.49	352	−0.45	0.971
DietGrp|WMED					
SympAvgb–SympAvgm1	7.71	3.58	352	2.15	0.140
SympAvgb–SympAvgm2	8.88	3.82	352	2.33	0.094
SympAvgb–SympAvgm3	8.71	3.87	352	2.25	0.112
SympAvgm1–SympAvgm2	1.18	1.29	352	0.91	0.798
SympAvgm1–SympAvgm3	1.00	1.65	352	0.61	0.930
SympAvgm2–SympAvgm3	−0.18	1.05	352	−0.17	0.998
DietGrp|WPD					
SympAvgb–SympAvgm1	10.87	0.93	352	11.72	<0.001
SympAvgb–SympAvgm2	13.15	0.99	352	13.32	<0.001
SympAvgb–SympAvgm3	13.21	1.00	352	13.20	<0.001
SympAvgm1–SympAvgm2	2.28	0.33	352	6.84	<0.001
SympAvgm1–SympAvgm3	2.34	0.43	352	5.50	<0.001
SympAvgm2–SympAvgm3	0.06	0.27	352	0.23	0.996

Note. Tukey comparisons were used to test the differences in estimated marginal means. Symptom Score average from baseline to month 3.

## Data Availability

The data that support the findings of this study are available from Grace-Farfaglia, Patricia (2021), “Self-Reported Diet and Health Outcomes of Participants of the CCSVI-Tracking Survey Study”, Mendeley Data, v1, http://dx.doi.org/10.17632/s6gzrnptsx.1.

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
