# Peer review of "Self-Reported Diet and Health Outcomes of Participants of the CCSVI-Tracking Survey Study"

_nutrients, 2021, doi:10.3390/nu13061891_

Round 1

Reviewer 1 Report

  1. Title of this study needs to be modified to better reflect study results. Something like following: “Self-reported diet and health outcomes of participants of the CCSVI survey study”
  2. In the introduction, author should mention results of dietary intervention studies in MS. There are several published studies suggesting association and beneficial effects of various diets (Swank, Wahls, McDougall) on MS symptoms. Results of these studies can help establish important of dietary choices in pwMS.
  3. Methods section need to clarify several points 1) is it cross-sectional analysis or longitudinal? If cross-sectional, which time-point data was used, if longitudinal then clarify the timeline. It is quite confusing to follow. 2) Did composite symptom score include quality of life scores? 3) Were EDSS and 6MTWT assessed by trained professional? If so, what are time-points? 4) Describe the diets here. Were participants given any description of diets or just names to choose from? 5) Statistical methods section needs to be revised. Please clarify purpose of different statistical tests. Seems both tests for association and differences between and within groups were performed.
  4. Results: Needs major revision. 
  5. Table 3 says symptoms average of 16 domains but method section reports 14 functional areas? What does baseline implies- how long before treatment?
  6. 1.1. MS diet adoption- analyses here is not needed to achieve study goals.
  7. 1.2- Results of one way ANOVA reported in line 178-79 need detailed description, p-values for between group differences for health outcomes. Results reported in line 188-199 are unnecessary and do not support aim of this study.
  8. Figure 2 and 3- what are p-values for significant change over time.
  9. 1.4- Seems unnecessary to report. Age, gender, disability etc. should be used as control variables in regression analysis. It is already known that these variables affect QOL.
  10. Discussion: Overall results show some effect of Mediterranean diet and need to report on previous literature on Mediterranean diet comparing with results of current study.

Reviewer 2 Report

In this MS, Author Dr Patricia Grace-Farfaglia reports detailed statistical analyses performed on MS patient-reported diet and health data from the CCSVI tracking study.

As far as association between dietary choices and health of MS suffering patients is concerned, the MS deals about a subject of great interest.

The MS is very readable and reports important analyses from a very capable research professional.

That said, I have few points that should be considered:

  • I have a general concern regarding the abstract. Readers may read just the abstract. Pls rephrase some part to make the message clearer. For example, author wrote about “MS diet”, what is it? Author wrote abbreviations such “CCSVI Tracking Survey” which is not understandable when reading solely the abstract.
  • Abstract line 12. “Adherence to the MS diet was not associated greater quality of life” term with seems missing.
  • In the abstract: Author wrote “A decline of symptoms was observed in all diet… with the most dramatic decline observed in participants from the Eastern Mediterranean diet region.” Is dramatic an appropriated term here? I am not a native English reader, but I am not sure dramatic is an appropriate term to define a decrease in painful MS symptoms.
  • In the abstract, in the sentence “The main effect for the within-subjects factor”, what is the within-subjects factor?
  • Lane 96 “Quality of Life (100 being perfect, 0 absence of MS symptoms)” Please check, I think there is a mistake.
  • I notice the use of “gender” term. I thought sex was the correct term to use in Science to refer to chromosomic equipment of an individual.
  • Concerning table 2: Author wrote “The EDSS categories: PPMS = Primary Progressive, SPMS = Secondary Progressive, RRMS=Relapsing-Remitting, PRMS = Progressive Relapsing.”. Actually EDSS is a score comprised between 0 and 10. The 4 mentioned “categories” are 4 types of MS pathology. I have to confess I do not know the PRMS category. Could the Author give detail on this category of MS pathology?
  • Figure 1 should be improved as it is not easy to find out the correspondence between each county and the portion in the graph.
  • Lane 365 The sentence “The PREDIMED study demonstrated that habitual intake of the MedDiet results in [36].” seems missing words.
  • In table 3, I do not understand the maximum value for EDSS (0.9) and the minimum value for 6MTWT baseline (0)?
  • Table 4. The table would be more readable if order of diets is the same for each variable.
  • Lane 379 “pwMS who early adopters of the PTA procedure.” Pls check this sentence.
